# Role of Mitochondria in Inflammatory Bowel Diseases: A Systematic Review

**DOI:** 10.3390/ijms242317124

**Published:** 2023-12-04

**Authors:** María José Sánchez-Quintero, Cristina Rodríguez-Díaz, Francisco J. Rodríguez-González, Alejandra Fernández-Castañer, Eduardo García-Fuentes, Carlos López-Gómez

**Affiliations:** 1Instituto de Investigación Biomédica de Málaga y Plataforma en Nanomedicina-IBIMA Plataforma BIONAND, 29590 Málaga, Spain; mj.sanchez@ibima.eu (M.J.S.-Q.); cris.rdrz@gmail.com (C.R.-D.); sandrus1978@gmail.com (A.F.-C.); 2Unidad de Gestión Clínica Cardiología y Cirugía Cardiovascular, Hospital Universitario Virgen de la Victoria, 29010 Málaga, Spain; 3Centro de Investigación Biomédica en Red de Enfermedades Cardiovasculares (CIBERCV), Instituto de Salud Carlos III, 28029 Madrid, Spain; 4Unidad de Gestión Clínica de Aparato Digestivo, Hospital Universitario Virgen de la Victoria, 29010 Málaga, Spain; 5Centro de Investigación Biomédica en Red de Enfermedades Hepáticas y Digestivas (CIBEREHD), Instituto de Salud Carlos III, 28029 Madrid, Spain

**Keywords:** inflammatory bowel diseases, Crohn’s disease, ulcerative colitis, mitochondria, oxidative stress

## Abstract

Mitochondria are key cellular organelles whose main function is maintaining cell bioenergetics by producing ATP through oxidative phosphorylation. However, mitochondria are involved in a much higher number of cellular processes. Mitochondria are the home of key metabolic pathways like the tricarboxylic acid cycle and β-oxidation of fatty acids, as well as biosynthetic pathways of key products like nucleotides and amino acids, the control of the redox balance of the cell and detoxifying the cell from H_2_S and NH_3_. This plethora of critical functions within the cell is the reason mitochondrial function is involved in several complex disorders (apart from pure mitochondrial disorders), among them inflammatory bowel diseases (IBD). IBD are a group of chronic, inflammatory disorders of the gut, mainly composed of ulcerative colitis and Crohn’s disease. In this review, we present the current knowledge regarding the impact of mitochondrial dysfunction in the context of IBD. The role of mitochondria in both intestinal mucosa and immune cell populations are discussed, as well as the role of mitochondrial function in mechanisms like mucosal repair, the microbiota– and brain–gut axes and the development of colitis-associated colorectal cancer.

## 1. Introduction

Inflammatory bowel diseases (IBD) are a group of chronic, inflammatory disorders of the digestive system, including Crohn’s disease (CD), ulcerative colitis (UC) and indeterminate colitis. Currently, IBD prevalence in Western countries is estimated to be over 0.3%, while newly industrialized countries show a lower but increasing prevalence of IBD [1]. Apart from the impact on the quality of life of patients, IBD have a huge economic impact on health systems. Therefore, therapies aimed at controlling disease progression to prevent hospitalizations, surgeries and work leaves are needed. Currently, an array of new therapies (Upadacitinib, Tofacitinib, Risankizumab, Ustekinumab) are joining first-line therapies (aminosalicylates, immune suppressors and anti-TNF antibodies). However, all these therapies are aimed at controlling the inflammation, and none of the therapies are aimed at targeting other mechanisms like reversion of fibrosis or mucosal repair. In order to design therapies targeting these alternative disease mechanisms and potentially improving the quality of life of patients, there is a need to better understand the pathophysiology of the disease and the cellular and molecular mechanisms underlying IBD. In this sense, mitochondrial biology may play a critical role, since the involvement of mitochondria in IBD has been described. In the present review, we aim to present the current knowledge on the role of mitochondria in IBD, focusing on how this knowledge may contribute to the design of new therapies targeting different disease mechanisms, not only inflammation but also reversion of fibrosis, mucosal repair, preservation of intestinal barrier integrity and prevention of dysbiosis of the intestinal microbiota.

## 2. Materials and Methods

The search was performed on PubMed on 19 September 2023, using the following terms: Mitochondria AND (Inflammatory Bowel Diseases OR Crohn’s Disease OR Ulcerative Colitis). A new search was performed on 7 November 2023 to include new articles published during the writing process. The results from both searches were further evaluated for inclusion. Reviews, editorials/comments or meeting reports, as well as articles in languages other than English, were discarded. Articles that were not published under an “open access” policy or were not accessible through the Virtual Library of the Andalusian Public Health System (BV-SSPA, https://bvsspa.es/, accessed on 19 September 2023) were also discarded. Finally, articles whose topic was not relevant to this review were discarded after reading the abstract or whole article.

## 3. Results and Discussion 

### 3.1. Literature Search

The search on PubMed yielded 268 articles. From those, one article was discarded for being retracted, fifty-five were discarded for being review articles, five were discarded for being editorials, comment articles or meeting reports, fourteen were discarded for being in languages other than English, twenty-nine were discarded for not being accessible through the BV-SSPA, and forty-seven were discarded after reading for not being relevant to the topic of this review. Particularly, articles citing only the mitochondrial pathway of apoptosis and not focusing on mitochondrial function were discarded. Thus, a total of 117 articles were used for the generation of this review. Literature search process is summarized in Figure 1.

### 3.2. Omics Signatures as Evidence of the Role of Mitochondria in IBD

Genome-wide association studies (GWAS) have increased our understanding of the molecular mechanisms underlying IBD. For instance, polymorphisms within the carnitine transporter genes OCTN1 and OCTN2 were found to be associated with CD [2]. Carnitine (and subsequently OCTN1 and OCTN2) are essential for the transport of long-chain fatty acids into the mitochondria for its β-oxidation. In a study focusing on the role of these transporters, the authors assessed the gut of the *Octn2*^−/−^ mice, which showed spontaneous atrophy of the intestine and colon as well as inflammation [2]. Ulcers, perforation and abscesses were present in most parts of the gut. This study showed the critical role of mitochondrial metabolism and bioenergetics for the normal function of the gut.

Apart from IBD-associated genes directly involved in mitochondrial function, other associated genes with a less-evident function in mitochondria have been found. An interesting study assessing the association of non-coding variants found in GWAS with gene enhancers found that the IBD risk variant rs1250566, previously annotated within an intron in the gene ZMIZ1, overlapped with an enhancer with a strong effect on the gene PPIF [3]. PPIF encodes cyclophilin D, a protein that is part of the mitochondrial permeability transition pore in the inner mitochondrial membrane and that controls the mitochondrial membrane potential. Other authors have focused on the study of genetic variants within the mitochondrial DNA, since GWAS typically focused on variants within the nuclear DNA [4]. These authors found an association of the variant 11,719 A/G (rs2853495) in the *MT-ND4* gene (a subunit of the mitochondrial Complex I) with UC specifically in male patients [4]. These results not only highlight the role of mitochondrial respiration in the pathophysiology of UC, but also suggest a role of mitochondrial DNA variants in sex-specific differences, like susceptibility or response to treatments.

Transcriptomics studies have also contributed to understanding the role of mitochondria in IBD. In a study performing RNA-Seq analysis of rectal samples from pediatric and adult UC, the authors found a robust decrease in mitochondria, aerobic tricarboxylic acid (TCA) cycle and metabolic functions in samples from IBD patients compared to controls [5]. Both mitochondrial-encoded and nuclear-encoded mitochondrial genes were down-regulated. In fact, all 13 mitochondrial-encoded genes from the respiratory chain complexes were down-regulated in samples from UC patients. Defects in mitochondrial function in mucosa were further confirmed (decreased Complex I activity in rectal biopsies from active UC, reduced membrane potential in colon biopsies from UC patients and reduced expression of the master regulator of mitochondrial biogenesis, PPARG coactivator 1 alpha (PGC1α), in rectal samples from UC patients). A more recent study comparing transcriptional profiles of colon mucosa from treatment-naïve IBD patients observed that inflamed tissue shows an overrepresentation of genes involved in inflammation and underrepresentation of genes involved in mitochondrial respiration, both in CD and UC [6]. These two transcriptomic studies support a role of mitochondrial biology in the development of IBD.

In a proteomics study using Rhesus macaques treated with dextran sodium sulfate (DSS) as an experimental model of colitis, a number of proteins involved in mitochondrial function were down-regulated in colon tissue compared to the protein levels in non-treated animals [7]. Oxidative phosphorylation, TCA cycle and fatty acid metabolism were down-regulated pathways involved in mitochondrial function. Another proteomics study used mitochondrial extracts from colon tissue from DSS-treated mice and found that the ferrireductase STEAP4 was selectively increased in treated animals [8]. Furthermore, STEAP4 was induced as soon as 3 days after treatment with DSS, which suggests a role of iron metabolism in the early stage of colitis. The authors further demonstrate that high levels of STEAP4 alter mitochondrial iron homeostasis and promote colitis-associated colorectal cancer.

Another study used proteomic analysis to assess lysine acetylation in a DSS-induced colitis mouse model and found that most differentially acetylated proteins were located in the cytoplasm and mitochondria [9]. KEGG pathways analysis revealed an enrichment of proteins involved in metabolism (tricarboxylic acid cycle, β-oxidation, oxidative phosphorylation) and protein processing within the endoplasmic reticulum. Colonic expression of the deacetylases SIRT3 and SIRT5 was up-regulated and likely is the reason mitochondrial and cytoplasmic proteins were deacetylated in DSS-treated mice. Interestingly, activity of SIRT deacetylases is known to be influenced by intracellular NAD^+^. A recent study showed reduced gene and protein expression of SIRT1 in DSS-treated mice, as well as reduced levels of NAD^+^ [10]. The authors previously observed higher levels of acetylated (inactive) PGC1α and mitochondrial dysfunction in DSS-treated mice [11], and concluded that decreased levels of NAD^+^ in intestinal mucosa are responsible for reduced SIRT1 and PGC1α levels, as well as for mitochondrial dysfunction. This was confirmed in a metabolomic study on the sigmoid colon of UC patients and healthy controls, where they studied the association of mitochondrial dysfunction with inflammation in IBD, as well as with reduced NAD^+^ levels [12]. The authors showed that inflamed tissues were distinct from both non-inflamed tissues from UC patients and tissues from healthy controls, with nicotinate and nicotinamide metabolism as the main pathway distinguishing these two groups. Thus, in agreement with a recent study [10], the authors observed that inflamed tissues have reduced levels of NAD^+^ and increased levels of its degradation products, NAM and ADPr. In addition, inflamed tissues showed a significant reduction in mitochondrial density and number compared to non-inflamed tissues from UC patients.

Finally, a study analyzing gene expression datasets and focusing on cuproptosis-related genes found differential expression of several genes involved in cuproptosis in CD, UC, coeliac disease and IBD-associated CRC [13]. Furthermore, the differential expression of these genes was closely related to mitochondrial respiration, tricarboxylic acid cycle and glycolysis. Genes and proteins associated to IBD and cited in this section are summarized in Table 1.

### 3.3. Role of Mitochondria within Intestinal Mucosa in the Development of IBD and Pathological Mechanisms

One of the very first studies to assess mitochondrial function in the context of IBD was the study by O’Morain et al. [14]. Unfortunately, the study only focused on the assessment of malate dehydrogenase as a marker of mitochondrial function and found no association with IBD. Nevertheless, a plethora of new studies has associated mitochondrial function with IBD, shedding light on the molecular mechanisms of this group of disorders. Alteration of the respiratory chain complexes and defective oxidative phosphorylation is only one of the many mechanisms linking mitochondrial function within the intestinal mucosa with IBD. These mechanisms are frequently interconnected, for instance, impaired oxidative phosphorylation leads to increased mitochondrial ROS production and oxidative stress, which subsequently alter the intestinal barrier. Furthermore, mitochondrial ROS activate the response to stress pathways, which are also important for mitochondrial function during mitochondrial stress. However, to help the reader to understand these mechanisms individually, we will present each of these mechanisms separately.

#### 3.3.1. Defects in the Oxidative Phosphorylation and Mitochondrial Respiration

A role of mitochondrial function in IBD has been long suggested. One of the first studies reporting a link between mitochondrial function and IBD was a case report that showed the case of a pediatric patient who presented with muscle weakness, seizures and respiratory insufficiency beginning at age 1 [15]. This patient presented metastatic Crohn´s disease at age 8 and responded to anti-TNF antibody therapy. Assessment of the mitochondrial function of a muscle specimen revealed functional defects in Complexes III and IV, suggesting that mitochondrial dysfunction was involved in the development of CD in this patient. Similarly, in a study assessing the ultrastructure of interstitial cells of Cajal, the authors observed that interstitial cells of Cajal from CD patients showed patchy alterations, including mitochondrial swelling and vacuolization [16]. Since the publication of these first studies, several other studies have deepened into the levels and activities of mitochondrial complexes in IBD patients and animal models. Related to mitochondrial enzyme activity, an interesting study observed between 50 and 60% decreased activity of mitochondria Complexes II, III and IV in the colonic mucosa of UC patients [17]. Another study of the levels of oxidative phosphorylation complexes in adults and pediatric IBD patients also observed decreased protein expression of Complex II in terminal ileum from CD patients and a reduction in Complexes I, II and IV in the rectum from UC patients [18]. When combining CD and UC patients, the expression of Complexes I, II and V was decreased in the ascending colon. A different study of the activity of mitochondrial electron transport chain complexes unraveled a decreased Complex II activity in colonic biopsies from UC patients compared to controls [19]. Interestingly, DSS-induce colitis mice also showed decreased activity of Complexes II and IV after DSS treatment. This dysfunction of the electron transport chain was also accompanied by increased nitrite levels and protein carbonyls, confirming increased oxidative stress. However, the samples did not show differences in complexes’ abundance, suggesting that decreased activity is caused by defective function of the complexes. A similar study observed opposite results, showing increased Complex IV activity in DSS-treated mice and no changes in Complex II activity [20]. The authors failed to detect increased protein carbonyls as a sign of oxidative stress upon treatment with DSS. However, increased levels of thiobarbituric acid-reactive substances (malondialdehyde) were detected, suggesting increased oxidative stress. Treatment with N-acetylcysteine prevented increased Complex IV activity and levels of malondialdehyde and reduced tissue damage induced by DSS. An interesting in vivo study using several T cell-mediated colitis mouse models showed that pathogenic T effector cells directly down-regulate the levels of succinate dehydrogenase/Complex II, but have no effect on the activity of Complexes I, III and IV [21]. Accumulation of succinate led not only to defective oxidative phosphorylation, but also to an inability to shift to glycolysis. Finally, a recent study using the indomethacin-induced colitis rat model and co-treatment with MitoTEMPO, a mitochondria-targeted antioxidant, observed that male and female rats differ in their mitochondrial profile [22], with females showing decreased intact colon mitochondrial respiration, colon mitochondria respiratory control ratio (RCR), Complex I activity and Complex IV activity upon treatment with indomethacin. However, female rats showed a better response to MitoTEMPO compared to male rats. Reduced catalase activity in male rats was the most plausible explanation for the increased mitochondrial ROS observed in both sexes after indomethacin treatment. This study proposes sex-specific differential mitochondrial function as one of the factors leading to a more severe disease course frequently observed in female IBD patients.

A novel study described a mechanism that impairs mitochondrial respiration and involves long non-coding RNAs [23]. Specifically, decreased expression of *GATA6-AS1* was observed in gut epithelia from IBD patients. Interestingly, the *GATA6-AS1* interactome is enriched in proteins involved in metabolism, tricarboxylic acid cycle and mitochondrial function. Moreover, it has been shown that *GATA6-AS1* silencing lead to the induction of TGM2, decreased the mitochondrial membrane potential and increased mitochondrial ROS production in the Caco-2 cell line. In fact, induction of TGM2 led to a decreased mitochondrial respiratory profile and reduced tricarboxylic acid cycle metabolites.

#### 3.3.2. Mitochondrial ROS and Oxidative Stress

Generation of ROS is one of the hallmarks of mitochondrial dysfunction associated with IBD, and generation of NADPH within mitochondria is essential for the redox balance. Isocitrate dehydrogenase (IDH2) is an enzyme that produces NADPH from NADP+ in mitochondria through the conversion of isocitrate to α-ketoglutarate. A study observed that knock-out of *Idh2* in mice aggravated dramatically the signs of DSS-induced colitis, increased the apoptosis of intestinal epithelial cells and increased oxidative stress [24]. Interestingly, treatment with the antioxidant N-acetylcysteine ameliorated the signs of colitis, apoptosis and oxidative stress, highlighting the important role of ROS in the pathophysiology of IBD. A study of Rubicon further discusses the role of mitochondrial ROS in IBD [25]. Rubicon is a positive regulator of the NADPH oxidase complex and it is involved in autophagy and inflammatory responses. The authors first demonstrated that Rubicon localizes in mitochondria and interacts with p22phox in the outer mitochondrial membrane of murine bone marrow-derived macrophages. More importantly, after stimulation with LPS, Rubicon over-expressing macrophages showed an increase in both total ROS and mitochondrial ROS and a down-regulation of subunits of the mitochondrial Complex III, suggesting decreased mitochondrial activity. Seahorse analysis showed that Rubicon-deficient macrophages tended to have more oxidative metabolism, while over-expressing Rubicon macrophages shifted towards glycolysis. Rubicon-deficient macrophages also had more mitochondrial mass and ATP generation and their mitochondrial dynamics were shifted towards mitochondrial fusion, while over-expressing Rubicon macrophages showed the opposite pattern. Mito-TIPTP, a designed inhibitor of the interaction of Rubicon and p22phox, enhanced the mitochondrial function. This molecule also prevented histological damage in the colon, prevented weight loss and increased survival of DSS-treated mice. Furthermore, the expression of both Rubicon and p22phox were increased in the colonic mucosa of UC patients compared to healthy controls, as well as the interaction of Rubicon with p22phox. This study highlights the importance of the mitochondrial homeostasis in colonic mucosa.

Some molecules have been reported to counteract oxidative stress in IBD. Prohibitin is a highly conserved protein that is expressed ubiquitously and localizes in the mitochondria of epithelial cells. This protein is a major component of mitochondrial inner membrane, interacting with Complexes I and IV of the electron transport chain. Expression of prohibitin is decreased both in colonic mucosal biopsies from Crohn’s disease patients and in the colon of two different experimental colitis mouse models (IL-10^−/−^ and DSS) [26]. Interestingly, the abundance of prohibitin is down-regulated by oxidative stress, and inversely, over-expression of prohibitin induces expression of GSH, an important antioxidant scavenger. Thus, prohibitin modulates the GSH/glutathione S-transferase antioxidant system in intestinal epithelial cells and prevents intestinal permeability induced by oxidative stress. In fact, prohibitin knock-down cells have increased autophagy via up-regulation of ROS [27]. The important role of prohibitin is further demonstrated by the fact that gene silencing of prohibitin reduces cell viability. Furthermore, cell viability is reduced when autophagy is inhibited with Baf A and further reduced in the presence of TNFα, all of which occur in inflamed colons in IBD.

Further studies showed that prohibitin interacts with STAT3 in mitochondria, more specifically, with phospho-S727-STAT3 [28]. Apart from its role as a transcription factor in the nucleus, STAT3 is known to also localize in mitochondria, where it interacts with Complexes I and II and regulates cellular respiration and metabolic function. The authors of this study used treatment with TNFα as a mitochondrial stressor (decreases the activity of ETC Complex I, decreases ATP production and increases mitochondrial-derived ROS). Thus, the authors observed that TNFα treatment down-regulates the abundance of prohibitin and its interaction with STAT3. Over-expression of prohibitin prevented mitochondrial dysfunction induced by TNFα exclusively in the presence of p-S727-STAT3, but not when p-S727-STAT3 expression was inhibited. The protective role of prohibitin seems to be even more critical in Paneth cells [29]. In that study, Jackson et al. observed that specific deletion of prohibitin in intestinal epithelial cells leads to spontaneous ileal inflammation, showing mitochondrial dysfunction in different cell types, but early defects in Paneth cells. More importantly, deletion of prohibitin specifically in Paneth cells was sufficient to cause spontaneous ileitis. Interestingly, the antioxidant MitoTEMPO ameliorated mitochondrial dysfunction, causing defects in Paneth cells and ileitis, suggesting mitochondrial dysfunction as the causative agent. Another study showed that targeting mitochondrial ROS may be a valid therapeutic approach based on the transcriptomic shift induced by MitoTEMPO in biopsy specimens [30]. The authors first showed that mitochondrial damage was present in many cell types (Paneth cells, goblet cells, enterocytes) in inflamed tissues. However, they concluded that mitochondrial damage was not the consequence of inflammation, since it was also present in dysfunctional Paneth cells (Type I) from non-inflamed tissues. Treatment with MitoTEMPO had consequences beyond the reduction in oxidative stress, as it normalized the expression of IL-17/IL-23, lipid metabolism and apoptotic gene signatures. These results suggest that, rather than be a consequence of inflammation, mitochondrial damage and oxidative stress can act as triggers of the inflammation.

Similar to MitoTEMPO, MitoQ is an orally available mitochondria-targeted antioxidant based on ubiquinone. MitoQ has been proven to ameliorate DSS-induced colitis via preserving mitochondrial morphology and reducing oxidative stress (reduced levels of malondialdehyde and nitrite damage) [31]. Interestingly, as in the previous study, effects on the immune system were observed upon a reduction in oxidative stress. More specifically, MitoQ prevented NLRP3 inflammasome activation and subsequently led to reduced IL-1β and IL-18 release in colons from DSS-induced colitis mice compared to untreated DSS-induced colitis mice. The authors used a human macrophage cell line (THP-1) to confirm that, while IL-1β and IL-18 release is induced upon treatment with H_2_O_2_ and ATP, co-treatment with MitoQ reduced the release of these two pro-inflammatory cytokines in a dose-dependent manner.

Another mitochondrial-targeted antioxidant, SkQ1, has been evaluated in models of colitis [32]. Fedorov et al. observed that SkQ1 prevented weight loss and rectal bleeding, reduced colon shortening and improved barrier function in the DSS-induced colitis mouse model. Treatment with SkQ1 also reduced the expression of pro-inflammatory genes (TNFα, IL-6, IL-1 and ICAM-1). Unfortunately, the authors failed to describe the effects of SkQ1 on the mitochondrial function of DSS-treated mice.

A recently described novel mechanism by which oxidative stress induces mitochondrial dysfunction is through the alteration of endoplasmic reticulum–mitochondrial associated membrane (MAM) [33]. This study showed that advanced protein oxidation protein products, indicative of oxidative stress, are enriched in active lesions from CD patients and positively correlated with markers of endoplasmic reticulum stress like GRP78, CHOP and IRE1α. Notably, the expression of CHOP has also been associated with the mitochondrial unfolded protein response. Mice treated with advanced protein oxidation products showed signs of intestinal inflammation, defective Paneth cell function (reduced Paneth cell number, reduced expression of Lyz and defensin proteins) and increased ER stress. More importantly, ER stress led to the alteration of MAM and subsequent mitochondrial dysfunction (mitochondrial swelling, reduced ATP production, mitochondrial Ca^2+^ overloading and aberrant MPTP opening). Interestingly, Paneth cell defects, MAM alteration and mitochondrial dysfunction could be prevented by treatment with TUDCA, an inhibitor of ER stress. The most important achievement of this study was that they showed the interlink between oxidative stress and ER stress and its consequences for mitochondrial and Paneth cell function.

#### 3.3.3. Mitochondrial Response to Stress

Both ROS and alteration of mitochondrial morphology represent stressors that activate specific stress responses. Particularly, the mitochondrial unfolded protein response (mtUPR) has been implicated in the pathology of IBD [34]. mtUPR induces phosphorylation of eIF2α via AP1/double-stranded-RNA-activated protein kinase (PKR) and further expression of mtUPR players like CPN60. Interestingly, *Pkr*^−/−^ mice, which do not show any specific phenotype, failed to induce PKR and CPN60 expression after treatment with DSS. Knock-down of *Pkr* prevented most of the DSS-induced signs of colitis, like weight loss, rectal bleeding and disease activity, suggesting that mtUPR is involved in the pathological mechanism. Rath et al. showed that both PKR and CPN60 are induced under inflammatory conditions, and they are both activated in intestinal epithelial cells from UC and CD patients. Another study used a conditional mouse model with non-phosphorylatable Ser51Ala mutant eIF2α in intestinal epithelial cells [35]. The authors showed swollen mitochondria with disrupted cristae in Paneth cells from mutant mice, as well as defective UPR signaling and hyperactivated ER stress in ileal epithelial cells, which confirms the role of eIF2α in the early activation of UPR upon stress. Using a mouse model with impaired mtUPR in intestinal stem cells (Hsp60^Δ/ΔISC^), Khaloian et al. observed that stem cells experiencing mitochondrial dysfunction transit towards dysfunctional Paneth cells [36]. Using human and mice intestinal organoids, the authors showed that abrogation of oxidative phosphorylation (using oligomycin) has a profound impact in the functionality of Paneth cells, while abrogation of glycolysis (using dichloroacetate, targeting the pyruvate dehydrogenase complex) had minimal effects. Interestingly, the effects of mitochondrial dysfunction in intestinal stem cells seem to be reversible by reinforcing mitochondrial respiration, which may be a novel strategy for therapy.

In a cohort study on infantile-onset IBD, the authors found loss-of-function mutations in ankyrin repeat and zinc-finger domain-containing 1 (ANKZF1) in four out of thirteen patients [37]. ANKZF1 translocates to the mitochondria upon cellular stress to regulate the degradation of damaged, misfolded and ubiquitinated proteins. The authors used ANKZF1-knock-down yeast to show that a 60% reduction in ANKZF1 expression led to a loss of mitochondrial integrity and a decrease in mitochondrial respiration under conditions of cellular stress.

Phosphatidylinositol signaling is associated with gastrointestinal disorders and malignancies. Using a zebrafish model with deficiency in the de novo synthesis of phosphatidylinositol, Thakur et al. demonstrated that disruption of phosphatidylinositol signaling lead to abnormal intestinal architecture and mucosal pathology and inflammation similar to IBD [38]. The authors further observed activation of the endoplasmic reticulum unfolded protein response, extensive mitophagy and mitochondrial depletion.

#### 3.3.4. Intestinal Permeability

Intestinal permeability is one of the IBD hallmarks more frequently associated with mitochondrial dysfunction. A study on ileal surgical resections in Ussing chambers to assess mucosal permeability to ^51^Cr-EDTA showed that alteration of the intestinal barrier induced by sodium caprate was accompanied by mitochondrial swelling and decreased epithelial ATP content [39]. The results from this study suggest that mitochondrial dysfunction is a consequence of intestinal barrier disruption. However, other studies have placed mitochondria as the causative agent of the barrier disruption. For instance, a study observed that barrier dysfunction can be prevented by using the mitochondrial-targeted antioxidant MitoTEMPO [40]. Using intestinal epithelial cell lines, the authors showed that barrier disruption is mediated via intracellular ROS. MitoTEMPO was efficient not only in preserving the intestinal barrier, but in suppressing DSS-induced IL-8 expression. Furthermore, the authors showed that treatment with MitoTEMPO in DSS-induced colitis mice prevented barrier dysfunction and led to a less-severe disease. Other studies have focused on the role of Multi Drug Resistance 1 (MDR1), an ATP-dependent efflux transporter highly expressed in the colon [41]. Ho and collaborators observed that silencing of MDR1 increased barrier permeability in vitro. As opposed to IL-10-deficient mice (spontaneously developing colitis), colonic inflamed epithelium from *Mdr1a*-deficient mice showed an accumulation of damaged and degenerating mitochondria, leading to dysfunctional mitochondrial function and a subsequent increase in mitochondrial ROS production. Further induction of mitochondrial ROS (using rotenone) accelerated and aggravated colitis in mice, while inhibition of mitochondrial ROS (using MitoQ) attenuated the development of colitis. Moreover, depletion of *Mdr1a* led to higher expression levels of *Sod2*, likely as a compensatory mechanism to counteract the increased ROS production. The authors observed a negative correlation between *SOD2* and *MDR1* genes in humans and showed that a specific deficiency of SOD2 in mouse colon makes them more susceptible to DSS-induced colitis. Another study supporting mitochondrial dysfunction as a causative factor for intestinal barrier disruption focused on the creatine transporter CRT. This study showed that knock-down of CRT in T84 cells lead to an inability to increase mitochondrial respiration and caused the energetic demand to rely on glycolysis upon mitochondrial stress (treatment with oligomycin and FCCP) [42]. In contrast, control T84 cells increased both mitochondrial respiration and glycolysis. In addition, the authors found that knock-down of CRT led to leaky barrier function in both T84 cells and colon organoids from mice. Interestingly, patients with IBD showed reduced CRT levels in colon biopsies compared to healthy controls.

### 3.4. Mitochondrial Dynamics, Mitophagy and Mitochondrial Biogenesis in IBD

The balance between mitochondrial fusion, fission and mitophagy defines the morphology not only of isolated mitochondria, but also of the whole mitochondrial network within a cell or tissue, which is of critical importance for cellular bioenergetics. In fact, the mitochondrial network is specific within each cell type, reflecting the metabolic and bioenergetic needs of each cell. In this sense, the mitochondrial network of intestinal epithelial cells (as assessed with immunofluorescence of TOMM20) has been described to be a continuous chainmail-like pattern largely devoid of swollen regions [43]. More interestingly, mitochondria are more densely packed in the subnuclear area compared to the lumen-apposed supranuclear area. In the same study, Chojnacki et al. detected mitochondrial fragmentation in DSS-treated mice and colonic tissue from UC patients compared to wild-type mice and healthy controls, suggesting a correlation between mitochondrial fission and inflammation. These results are in line with two other studies associating higher levels of phosphorylated DRP1 and mitochondrial fission with inflammation [44] and mitochondrial fusion with mucosal repair [45]. Another study observed increased gene expression of mitochondrial dynamics proteins (Drp1, Fis1, Opa1, Mfn1 and Mfn2) and higher levels of phosphorylated DRP1 in DSS-treated mice [46], in agreement with previous studies. The authors further administered the specific inhibitor of mitochondrial fission P110, resulting in a reduction in disease severity (prevented shortening of colon length, reduced colon motility and yielded a decreased macroscopic disease activity score). Treatment with P110 partially abrogated mitochondrial fragmentation in DSS-induced colitis and improved mitochondrial respiration and short and long fatty acid oxidation. However, inhibition of mitochondrial fission by P110 did not affect other hallmarks of colitis like bacterial dysbiosis, epithelial permeability or cytokine production, suggesting that alteration of mitochondrial function is itself an independent mechanism in IBD.

Mitophagy is another important regulator of mitochondrial quality and mitochondrial mass. Mitophagy removes damaged mitochondria, thereby reducing the mitochondrial ROS and subsequently preventing the induction of the NLRP3 inflammasome by ROS. It has been observed that both the number of damaged mitochondria and mitophagy are increased in intestinal tissue from UC patients, and this correlates with disease severity [47]. In that study, the authors identified Heat Shock Transcription Factor 2 (HSF2) as a regulator of mitophagy through the PARL/PINK1/Parkin pathway. ATG16L1 is an important player of the autophagy machinery and polymorphisms in ATG16L1 have been repeatedly associated with IBD. It has been shown that intestinal organoids from mice lacking the *Atg16l1* gene in the colon contain a significantly higher number of aberrant, swollen mitochondria with a loss of cristae, as well as increased production of mitochondrial ROS [48]. These results highlight the relevant role of autophagy/mitophagy in preventing excessive production of ROS and its damaging effects. *IRGM* is another gene associated with IBD and involved in autophagy, as well as in cells’ defense against microorganisms. A study found that the IRGM protein localizes to mitochondria (likely the mitochondrial inner membrane or matrix). The study described that IRGM knock-down cells show abnormally elongated mitochondria, similar to DRP1 knock-down cells, suggesting that IRGM has a direct role in mitochondrial fission [49]. Moreover, the study showed that the splicing isoform IRGMd specifically binds to Cardiolipin and promotes not only mitochondrial fission and autophagy, but also mitochondrial depolarization and cell death. Interestingly, a study assessing the effect of *Irgm1* knock-down on the DSS-induced colitis mouse model confirmed the role of *Irgm1* on mitochondrial fission [50]. The authors observed that exaggerated intestinal inflammation in *Irgm1* knock-down mice was associated with impaired autophagy and mitophagy and the presence of dysfunctional elongated/tubular and swollen mitochondria. The effects of *Irgm1* knock-down were critical for the function of Paneth cells, which showed several abnormalities. Interestingly, the mouse model lacking *Atg16l1* in intestinal epithelial cells also showed a loss of Paneth cells [48], suggesting that this important cell type is particularly sensitive to impaired autophagy and dysfunctional mitochondrial function. NIX is another player of the autophagy/mitophagy machinery that seems to be involved in IBD [11]. Vincent et al. observed increased expression of NIX (indicating autophagy) and DRP1 (a common precursor of mitophagy, indicating mitochondrial fission) in UC patients. Furthermore, colocalization of NIX with cytochrome C suggested that mitophagy is enhanced in UC patients. Similar results were obtained with the DSS-induced colitis and the adoptive T cell transfer mouse models. The authors nicely showed that mitophagy was induced via the generation of ROS and subsequent activation of HIFα and that treatment with the mitochondrial-targeted antioxidant MitoTEMPO reduced NIX levels and attenuated colitis. Furthermore, knock-out mice lacking the NIX protein were more susceptible to DSS-induced colitis and accumulated dysfunctional mitochondria that produced higher levels of ROS. All these results point to the induction of mitophagy in IBD, likely as a consequence of mitochondrial damage, and highlight the importance of these mechanisms to preserve mitochondrial function during colitis. A study deepened the interlink between mitochondrial ROS and mitophagy using a conditional mouse model [51]. The authors observed that deficiency in oxidative phosphorylation (achieved with the tamoxifen-induced deletion of *Tfam*) sensitizes mice quiescent cells to oxidative stress. Interestingly, the abrogation of oxidative phosphorylation in quiescent cells led to suppression of autophagy. In vitro blocking of ROS with either N-acetyl cysteine or catalase over-expression yielded similar results in terms of suppression of autophagy, while treatment with low-dose H_2_O_2_ of oxphos-deficient cells rescued resistance to ROS. ATG4B activity, which can act as a double-edged sword by promoting and inhibiting autophagy, was found to be too elevated in oxphos-deficient cells and removes phosphatidylethanolamine from LC3B-II. In contrast, low ROS levels produced by oxidative phosphorylation seem to modulate ATG4B activity, thus contributing to the maintenance of a basal autophagic flux.

A recently described mechanism of mitophagy induction is via polystyrene nanoplastics [52]. Xu et al. observed that polystyrene nanoplastics induced Crohn´s ileitis-like signs in mice, showing a decreased villus length, increased crypt depth, increased mucus secretion, increased Th-17 cell infiltration and increased expression of IL-1β and TNFα in ileal lamina propria. Using both mice and Caco-2 cells, the authors determined that polystyrene nanoplastics induced necroptosis in intestinal epithelial cells, rather than apoptosis. Interestingly, polystyrene nanoplastics co-localized with mitochondria and induced increased mitochondrial ROS and mitochondrial membrane depolarization, together with an alteration of mitochondrial morphology (disrupted membrane, fuzzy cristae). Up-regulated expression of PINK1 and PARKIN and colocalization of LC3B puncta with mitochondria were clear signs of mitophagy activation subsequent to mitochondrial damage. However, the authors found that downstream mitophagy flux was blocked due to defective lysosomal function due to the accumulation of polystyrene nanoplastics in lysosomes, which further led to necroptosis. Together, these results nicely showed how an environmental contaminant can induce ileitis via alteration of mitochondrial function and mitophagy.

Finally, mitochondrial biogenesis, together with mitochondrial fusion/fission and mitophagy, contributes to the generation of functional mitochondrial mass within the cell. A study observed that SMYD5 negatively regulates post-transcriptionally the expression of PGC1α, the master regulator of mitochondrial biogenesis, and that depletion of SMYD5 protects mice from DSS-induced colitis [53]. SMYD5 was found to be up-regulated in IBD patients with active inflammation. In contrast, PGC1α was down-regulated in IBD patients compared to controls and showed an inverse gene expression pattern to that of SMYD5. Overall, SMYD5 seems to be promoting IBD progression by enhancing PGC-1α degradation and the subsequent impairment of mitochondrial function. A study assessing the effects of vitamin A supplementation on the TNBS rat model of colitis found an important involvement in mitochondrial function [54]. Particularly, the reduced colon damage and reduced myeloperoxidase activity (a surrogate marker of neutrophil infiltration) was associated with enhanced oxidative phosphorylation and increased expression of NRF1 and TFAM in colon tissues. Both NRF1 and TFAM are involved in mitochondrial DNA (mtDNA) transcription and replication and, therefore, in mitochondrial biogenesis. In fact, the authors also observed increased levels of mtDNA (used as a marker of mitochondrial mass) in colons from TNBS + vitamin A-treated rats compared to TNBS-treated rats.

In conclusion, mitochondrial fission/fusion, mitophagy and mitochondrial biogenesis contribute together to maintain a pool of healthy, functional mitochondria and are, therefore, critical mechanisms to deal with mitochondrial dysfunction during colitis. These mechanisms and genes previously described are summarized in Figure 2.

### 3.5. Involvement of Mitochondria in Mucosal Repair and Fibrosis

Fibrosis of the intestinal mucosa is one of the most concerning disease mechanisms in IBD. Fibrosis of the intestinal mucosa causes stenosis, which makes intestinal motility difficult and frequently leads to surgery. Thus, understanding the molecular and cellular pathways leading to fibrosis is critical to prevent IBD patients from requiring surgery. In one study assessing the role of vitamin D receptors in intestinal fibrosis [55], the authors demonstrated that vitamin D receptors inhibit fibrosis and that vitamin D modulates the progression to fibrosis in the DSS-induced colitis mouse model. Interestingly, the authors also observed that vitamin D was involved in barrier integrity by maintaining mitochondrial function. Furthermore, in colonic tissues from patients with CD two mitochondrial genes, VDAC1 and ATP5A, were down-regulated in stenotic areas, suggesting that mitochondrial dysfunction was involved in the progression to fibrosis.

Jurickova et al. used human intestinal organoids developed from induced pluripotent stem cells to study the effects of polymorphisms in the DUOX gene. Their results showed that both butyrate and eicosatetraynoic acid (ETYA) down-regulated the expression of genes involved in the generation of the extracellular matrix (ECM) [56]. In addition, butyrate inhibited ROS production upon toxin stimulation (*Pseudomonas* toxin pyocyanin). Both butyrate and ETYA suppressed the expression of genes involved in wound healing and ECM formation, while only chronic exposure to ETYA reduced tissue stiffness.

Mucosal healing is another goal for future therapeutics, but the mechanisms remain poorly described. To study mucosal healing, Lan et al. treated mice with 3.5% for 5 consecutive days [45]. The authors observed that inflammation peak at days 7–10. By day 13, the expression of markers of differentiated colonocytes like Atp1a1, Nhe1 and Muc2 was restored, suggesting spontaneous mucosal repair. During days 7–10, gene expression of *Pgc1α*, *Nrf2* and *Tfam* was down-regulated and the mitochondrial mass was decreased (as measured with immunostaining of TOMM-20). Mitochondrial swelling was also evident in the intestinal epithelium of DSS-treated mice by day 10. However, from day 13, gene expression of mitochondrial genes like *FoxO3* and *Mfn2*, as well as mitochondrial fusion, was up-regulated. Additionally, mitochondrial mass was restored and mitochondrial morphology appeared normal by day 21. Mucosal repair was also accompanied by an increase in mitochondrial respiratory spare capacity and mitochondrial ATP production. These results highlight the importance of mitochondrial bioenergetics in the process of mucosal repair. In a recent study, the authors observed a higher abundance of phosphorylated DRP1, a marker of mitochondrial fission, in samples from UC patients compared to healthy controls, as well as in mucosa from DSS-treated mice compared to untreated animals [44]. Interestingly, the results showed that inhibition of mitochondrial fission using P110 improved the recovery of DSS-treated mice. Further data showed that excessive mitochondrial fission impairs the metabolism of butyrate via ROS. In combination, these two studies show that a fine-tuned regulation of mitochondrial dynamics is necessary to provide the cell with the energy and metabolic requirements to achieve mucosal healing.

### 3.6. Role of Mitochondria in the Immune System

Different cells from the adaptive and innate immune system are involved in the pathogenesis of IBD, among them T cells, natural killer (NK) cells, macrophages and neutrophils. Different studies have observed a role of mitochondrial function in these three cell subpopulations in the context of IBD. For instance, mitochondrial function is described to have a role in the fate of T cells through the stimulation of interferon genes (STING). STING has been reported to transform proinflammatory IFNγ^+^ Th1 cells into less-pathogenic IL-10^+^ IFNγ^+^ Th1 through translocation of STAT3 to the nucleus and mitochondria [57]. In the nucleus, STAT3 activates *Blimp1* expression, which further induces the expression of IL-10. In mitochondria, activation of mito-STAT3 enhances mitochondrial oxidation. Interestingly, suppression of glucose oxidation or glutamine oxidation inhibits IL-10 production, but not suppression of fatty acid oxidation. These data highlight how the metabolic state can modulate the fate of T cells. Regulatory T cells (Tregs) are another T cell subpopulation with a key role in IBD, counteracting the pro-inflammatory response. Their metabolic state also has an impact on their function. A study using transmission electron microscopy revealed that naïve CD4^+^ T cells from peripheral blood of healthy donors have rounded mitochondria, while induced Tregs have elongated, tubular mitochondria associated with a more oxidative metabolism [58]. Induced Tregs also show a higher proportion of mitochondria- and ER-associated membranes, where IP3R1 and VDAC interact. TGF-β1 enhanced the proportion of this association. In fact, induced Tregs developed in the presence of TGF-β1 showed a higher oxygen consumption rate in seahorse than induced Tregs developed in the absence of TGF-β1. The authors further demonstrated that disrupting the entry of pyruvate into the mitochondria sensitizes Tregs to IL-12, so that they acquire a more effector phenotype. Similarly, treatment with IL-21 rewired the metabolism of Tregs to a more glycolytic state and dissociated mitochondria–ER contact sites.

Activation of the T cell receptor (TCR) is a key process in the activation of T cells. Upon TCR stimulation, Ca^2+^ is released from the endoplasmic reticulum. Mitochondrial uptake of cytosolic Ca^2+^ functions as a buffer, but it also activates oxidative phosphorylation, thus fulfilling two different roles of the mitochondria in activated T cells: to provide the ATP demanded by activated cells and to provide ROS, which will later act as a cytotoxic weapon of T cells. Furthermore, TCR activation also induces the release of ATP through pannexin-1 hemichannels [59]. This extracellular ATP acts later as an autocrine signal through P2X receptors, promoting MAPK signaling. These results show the importance of mitochondrial activity in the function of immune cells. Furthermore, treatment with oxidized ATP, an antagonist of P2XRs, inhibited the expression of IL-2. The authors further observed that treatment with oxidized ATP could ameliorate the symptoms in an IBD mouse model.

NK cells are also an important immune population involved in IBD. Zaiatz Bittencourt et al. published a study in which peripheral blood of healthy controls and IBD patients was analyzed. They observed that IL-12- and IL-15-stimulated NK cells from IBD patients had significantly lower production of IFNγ and reduced killing capacity compared to NK cells from healthy controls [60]. NK cells from IBD patients had, in addition, lower basal mitochondrial respiration and tended to be more glycolytic. Furthermore, upon cytokine stimulation, CD56^bright^ NK cells significantly up-regulate their mitochondrial mass and polarization compared to CD56^dim^ NK cells. However, CD56^bright^ NK cells from IBD patients showed impaired up-regulation of mitochondrial mass and mitochondrial membrane potential compared to healthy controls, showing that alteration of mitochondrial function in NK cells is present in IBD patients.

Similar to NK cells, the metabolic profile of activated macrophages is dependent on the stimuli. For instance, LPS drives macrophages towards glycolysis and a pro-inflammatory phenotype, while IL-4 drives macrophages towards oxidative phosphorylation and an anti-inflammatory phenotype. A study showed that IL-10-deficient macrophages stimulated with LPS have a reduced oxidative phosphorylation rate compared to LPS-induced wild-type macrophages [61]. This reduction was due to a loss of mitochondrial fitness. The authors observed that IL-10 prevents the accumulation of dysfunctional mitochondria by promoting autophagy and that IL-10-induced inhibition of mTOR signaling was responsible for both the preservation of mitochondrial integrity and the inhibition of inflammasome activation.

Macrophage metabolism and inflammatory responses are further interconnected through other mechanisms. A few years ago, Wolf et al. found that peptidoglycan induces activation of the NLRP3 inflammasome through the inhibition of hexokinase and release of hexokinase from the mitochondrial outer membrane [62]. Interestingly, glucose 6 phosphate and citrate (natural inhibitors of hexokinase) had the same effect on the activation of NLRP3 as peptidoglycan, which would be a link in the association of Western-style diets (rich in sugar) and IBD epidemiology. The interaction of hexokinase with VDAC at the outer mitochondrial membrane is involved in the regulation of glycolysis, mitochondrial stability, ROS production and permeability transition pore formation. Therefore, the release of hexokinase from the mitochondrial outer membrane would have a profound impact on mitochondrial metabolism. However, the authors observed that peptidoglycan induced a physiologically tolerable level of hexokinase release that did not involve the degradation of mitochondria or a collapse of total cellular mitochondrial function. Plant lectins are also reported to induce activation of the NLRP3 inflammasome in macrophages via an accumulation of misfolded proteins in the ER lumen and ER stress, which then promotes mitochondrial damage and mitochondrial ROS production [63]. Interestingly, the authors observed that the inhibition of Ca^2+^ release from the ER prevented mitochondrial damage and NLRP3 activation, suggesting that plant lectins can act as exogenous danger signals and activate the NLRP3 inflammasome through mitochondrial damage and the generation of ROS.

Neutrophils are an important contributor to inflammation in IBD, but also to disease resolution. Deletion of lymphotoxin β receptor in neutrophils is enough to induce weight loss, colon shortening and aggravated DSS-induced colitis in mice [64]. Riffelmacher et al. observed that a lack of lymphotoxin β receptors in neutrophils induces a transcriptional reprogramming leading to an alteration of mitochondrial function. Particularly, lymphotoxin β receptor-deficient neutrophils had a higher mitochondrial mass but a lower mitochondrial surface, suggesting a lower capacity for ATP production and more glycolytic activity. However, ROS production after stimulation was increased in lymphotoxin β receptor-deficient neutrophils and contributed to the more severe phenotype shown by mutant mice.

Noteworthy, the role of ROS in the pathogenesis of IBD is more complex than simply inducing cell damage. As such, ROS production and mitochondrial activity in the mucosa have a dual effect on the immune cells. A study used transgenic mice expressing the human IF1 gene in the intestine, which inhibits ATP synthase, and observed that production of mitochondrial ROS driven by partial inhibition of ATP synthase prevented tissue from developing DSS-induced colitis [65]. Consistent with the inhibition of ATP synthase, intestinal cells showed reduced mitochondrial basal respiration and a shift towards glycolysis. Maybe because of this metabolic shift, the increase in mitochondrial ROS was mild. Mitochondrial ROS are thought to play a role in the damaging of intestinal cells and inducing inflammation. However, Formentini et al. observed an anti-inflammatory profile compared to wild-type mice, with increased expression of anti-inflammatory cytokines and a reduction in several pro-inflammatory cytokines. Moreover, macrophages shift from pro-inflammatory M1 in DSS-treated wild-type to anti-inflammatory M2 in DSS-treated transgenic mice. This shift in macrophage polarization can be blocked by blocking mitochondrial ROS, proving the involvement of mitochondrial ROS in modulating the immune system. The authors further discuss the opposite role that low (beneficial) and high (damaging) mitochondrial ROS levels have on tissue integrity and immune system modulation. In fact, the intestinal epithelial cells can also be considered an important part of the innate immune system, as they act as a barrier against microbes. As such, dysregulation of mitochondrial homeostasis also has an effect on the immune function of these cells. Pascual-Itoiz et al. used DSS-treated mice deficient in the methylation-controlled J protein (MCJ), a natural inhibitor of mitochondrial Complex I [66]. Mice with a deficiency in MCJ had a worse histology outcome after treatment with DSS. *Mcj*-deficient mice treated with DSS showed reduced neutrophil infiltration and higher TNF bound to membranes in macrophages. However, the lack of MCJ did not result in a dramatically altered innate immune cell population. In contrast, cytokine production in the colon mucosa was dysregulated in the absence of MCJ. More specifically, the lack of MCJ also led to a higher production of proinflammatory genes like *Nos2*, *Il1b*, *Tgfb*, *Il10 Tnfα*, *Tlr9* and *MyD88*, which subsequently led to dysbiosis of the colonic microbiota.

Apart from ROS, mitochondria are the source of other elements with pro-inflammatory potential. The endosymbiotic origin of mitochondria causes many of its elements to be similar to bacterial components that can be recognized by pathogen-associated molecular pattern (PAMP) receptors like TLRs. Because of this, when elements like N-formyl peptides, mtDNA, ATP or Cardiolipin are exposed out of their mitochondrial context, they are considered danger-associated molecular patterns (DAMPs) and can trigger pro-inflammatory responses. In fact, it has been reported that patients with active UC or CD have higher levels of circulating free mtDNA and N-formyl peptides in plasma compared to matched healthy controls or irritable bowel disease controls [67]. Interestingly, mtDNA levels in plasma were even higher in those patients with severe UC or CD. The authors of this study further observed mtDNA levels in plasma from DSS-induced colitis mice and hypothesized that mitochondrial DAMPs are released after mitochondrial damage and enhance the inflammatory response. A later study observed that oligomerization of voltage-dependent anion channel 1 (VDAC1) mediates mtDNA movement across the outer mitochondrial membrane [68]. Interestingly, expression of VDAC1 is increased in colon tissue from IBD patients and DSS-treated mice. VBIT12, a specific interacting molecule of VDAC1 designed to prevent oligomerization of VDAC1, reduced the H_2_O_2_- or DSS-induced release of mtDNA in a mouse CRC cell line, CT-26. VBIT12 also inhibited DSS-induced VDAC1 over-expression, VDAC1 oligomerization, apoptosis, increased mtROS production and increased intracellular Ca^2+^ levels in CT-26 and Caco-2 cell lines. Furthermore, both VBIT12 and VBIT4 (another interacting molecule of VDAC1) ameliorated signs of colitis in the DSS- and TNBS-induced colitis mouse model. VBIT12 prevented the release of mtDNA, showing reduced serum levels in DSS-treated mice co-treated with VBIT12. As a consequence, many pro-inflammatory markers were reduced upon treatment with VBIT12 (activation of NF-κB, activation of NLRP3, expression of IL-1β, TNFα and IL-6, and levels of MPO and NO). Interestingly, the purinergic receptor P2X7, capable of recognizing ATP as a mtDAMP, was more highly expressed in the colon of DSS-treated mice than in controls and VBIT12-co-treated mice. All these results show that targeting the release of mtDAMPs is a valid therapeutic approach to reduce inflammation in colitis.

As stated before [30], modulating mitochondrial function has direct consequences not only in the intestinal mucosa, but also in the immune system. A study focusing on the mitochondrial translocator protein (TSPO) showed that targeting TSPO with specific drug ligands modulates the expression of CINC-1 (homolog to IL-8) and TNFα in plasma and colon tissue of the DSS-induced colitis rat model [69]. In normal intestinal mucosa, TSPO is localized in surface epithelial cells, but not in goblet cells. TSPO expression is enhanced in areas of low- and high-grade dysplasia adjacent to colon cancer, as well as in colon cancers. In inflamed tissues from IBD patients, TSPO expression is diffuse through all glands compared to the specific surface epithelia expression observed in normal mucosa. Overall, TSPO expression is increased in biopsies from IBD patients compared to healthy controls. In a more recent study, the authors observed that *Tspo* knock-out mice had a more severe disease upon treatment with DSS [70]. Interestingly, knock-out and wild-type mice showed a different modulation of genes involved in mast cell (cd34 and mcp6), macrophage (F4/80) and dendritic (cd11c) cell function, as well as TNFα. All these studies highlight how the alteration of mitochondrial homeostasis can act downstream to modulate the immune system.

The association of genes with IBD in GWAS has contributed to a better understand of the molecular mechanisms of IBD. This is the case for *LACC1*, a gene associated with CD. The LACC1 protein localizes to the cytoplasm and mitochondria and is involved in signaling through pattern-recognition receptors in macrophages [71]. Cells harboring the risk allele Val254 express lower levels of both pro- and anti-inflammatory cytokines. Lahiri et al. observed that both the costimulatory contribution of mitochondrial ROS in the activation of pattern-recognition receptors and the subsequent ROS production aimed at microbial clearance were altered in macrophages harboring the CD-risk allele. Another gene previously associated with CD is *FAMIN*, whose function remains unknown. In a groundbreaking study, Cader et al. revealed that FAMIN has several enzymatic activities involved in purine metabolism [72]. The authors further observed that this purine nucleotide cycle in macrophages is critical for the balance of mitochondrial oxidative phosphorylation and glycolysis, and it is of particular importance in M1 (pro-inflammatory) macrophages, which compromise glycolysis. This purine nucleotide cycle supplies mitochondria with fumarate, which synchronizes mitochondrial activity with glycolysis by balancing electron transfer into mitochondria. *CARD9* is one of the genes that have been more strongly associated with IBD, and recent studies of its role have unraveled the involvement of mitochondrial function in neutrophils [73]. The authors observed that deletion of *CARD9* in epithelial cells did not change the susceptibility to DSS-induced colitis, while deletion of *CARD9* in neutrophils aggravated colitis, with increased weight loss and histological score and decreased colon length. Increased expression of MPO and Lcn2 in the colon at day 12 and similar expression of Lcn2, Cxcr2 and S100A8 at day 9, when neutrophil recruitment was maximal, suggest that *CARD9*-deficient neutrophils are efficiently recruited to the inflamed tissue. However, the number of activated neutrophils in inflamed colon decreased. In fact, the killing capacities of neutrophils against *C. albicans* were strongly affected by *CARD9* deletion. The CARD9 protein is involved in autophagy, but defects in autophagy did not seem to be the mechanism behind the decreased killing capacity. In contrast, the authors observed increased apoptosis of neutrophils, which was caused by mitochondrial dysfunction. More specifically, mitochondria in *CARD9*-deficient neutrophils showed an over-activation of oxidative phosphorylation and decreased glycolytic activity, which normally is the main energy source of neutrophils. Subsequently, *CARD9*-deficient neutrophils produced a higher amount of mitochondrial ROS, likely leading to apoptosis.

### 3.7. Role of Mitochondria in Gut–Microbiota Axis

Alterations in the interaction of microbiota with the intestinal mucosa are recognized as a critical mechanism in the development of IBD. Mitochondrial function within epithelial cells plays, therefore, an important role in the maintenance of this balance. Almost 20 years ago, it was reported that a dysfunction of mitochondria in epithelial cells in combination with commensal, nonpathogenic bacteria (*E. coli*) induced barrier disruption, bacterial translocation, up-regulation of IL-8 and increased infiltration of immune cells [74]. Mitochondrial dysfunction in the host contributes, therefore, to barrier disruption and subsequently, to entry of bacteria into the mucosa. Invasive bacteria in epithelial cells are encapsulated and degraded in autophagosomes/autolysosomes. Importantly, many genes involved in autophagy are associated with IBD (*ATG16L1*, *NOD2*, *IRGM*). Thus, mitochondria and autophagy are key mechanisms in the microbiota–host interaction. Mitochondria are also a source for autophagosome membranes. TRIM31, a protein attached to the mitochondrial outer membrane, plays a critical role in providing PE from mitochondria to autophagosomes independently of the conventional Atg5–Atg12–Atg16 autophagy pathway [75]. Interestingly, the authors of this study observed that protein and gene expression of TRIM31 is severely down-regulated (45% and 85%, respectively) in ileum from CD patients. In another study, the authors assessed the combination of mitochondrial dysfunction and autophagy defects and its effects regarding the host–microbiota interaction [76]. The results showed that increased ROS (by treatment with the ionophore Dinitrophenol) promoted bacterial internalization via ERK1/2 phosphorylation, while defects in the *NOD2* gene lead to reduced killing of bacteria in autolysosomes. The authors hypothesized then that decreased barrier function due to dysfunctional mitochondrial is amplified by a lack of NOD2. In a transcriptomic study, Ruiz et al. first generated a panel of transcripts differentially regulated in low-ATP conditions, simulating a mitochondrial dysfunction [77]. Then, the authors observed that inflamed tissue from IBD patients was abundant in transcripts from this panel, which correlated with signatures of TLR4 and NOD2 signaling, suggesting an interconnection of bacterial signaling through TLR4 and NOD2 and mitochondrial dysfunction. Interestingly, it was previously shown that activation of endoplasmic reticulum stress counteracts the effect of dysfunctional mitochondria on barrier function by increasing the killing of invasive bacteria within autophagosomes [78]. Notably, it has been shown that epithelial mitochondria–microbiota interaction is a two-way street. An in vitro study using T84 monolayers and adherent invasive *E. coli* showed that the transcriptomic signature of T84 cells was altered by the infection with *E. coli*, with many genes involved in mitochondrial function being differentially expressed [79]. Furthermore, infection with the strain F82 induced swelling of mitochondrial cristae, loss of mitochondrial membrane potential and mitochondrial fragmentation. This study showed that the interlink between mucosal mitochondria and microbiota goes beyond the disruption of the intestinal barrier by dysfunctional mitochondria and that specific microbes can induce mitochondrial dysfunction without previous disruption of the barrier. A second study assessing the role of estrogen-related receptor alpha (ESRRA) also supports the bi-directional interlink between bacteria and epithelial mitochondria [80]. ESRRA is a transcription factor involved in many cellular mechanisms like mitochondrial biogenesis and autophagy. ESRRA knock-out mice are more susceptible to DSS-induced colitis and show damaged mitochondria and defects in autophagy, as well as dysbiosis. Interestingly, fecal microbiota transplantation from wild-type mice was enough to increase the survival of DSS-treated ESRRA-deficient mice and improved the autophagic flux. Similar results were observed in *Mcj*-deficient mice treated with DSS, where higher production of pro-inflammatory cytokines in the intestine and microbiota dysbiosis were associated with a higher susceptibility to DSS [66]. These results suggest that mitochondrial function in the intestinal mucosa can modulate intestinal microbiota, subsequently affecting the susceptibility to DSS. In a more recent study, Peña-Cearra et al. transplanted microbiota from *Mcj*-deficient mice treated with DSS to germ-free mice [81]. Germ-free mice colonized by microbiota from *Mcj*-deficient mice had a higher disease activity index upon treatment with DSS, as well as a higher expression of the pro-inflammatory cytokines IL-1β and Lcn2. In contrast, co-housing *Mcj*-deficient mice with wild-type mice reduced their disease activity score and histological score.

Invasion of epithelial cells in not the only way through which gut bacteria interact with the mucosa. The metabolism of commensal bacteria and the epithelial wall are closely interconnected and a number of studies highlight how the imbalance of one of them affects to the other. For instance, a study showed that butyrate, a short-chain fatty acid (SCFA) produced by intestinal microbiota, induces necrosis of MCE301 (murine normal colonic epithelial cells) and Caco-2 cells at 8 and 20 mM, respectively [82]. Mitochondrial swelling and decreased mitochondrial membrane potential likely mediate the necrosis induced by butyrate, while prednisolone and 5-aminosalicylic acid prevented necrosis in a dose-dependent manner. It remains unknown whether the concentrations at which the experiments were carried out are clinically significant. Nonetheless, this damaging effect of butyrate may be accentuated in a context of altered metabolism of butyrate. Butyrate is metabolized to acetyl CoA within mitochondria through five different enzymes. A study assessing the activity of three out of these five enzymes in colonic biopsies showed that mitochondrial acetoacetyl CoA thiolase activity was decreased specifically in UC patients, regardless of the inflammation status of the tissue assessed [83]. Rescue of thiolase activity β-mercaptoethanol suggested that critical sulphydryl residues of this enzyme are modified by ROS. In fact, Lewis et al. did not observe damaging effects of butyrate, but rather benefits in terms of increased integrity of the intestinal barrier [84]. More specifically, they observed that alteration of mitochondrial morphology and activity induced by treatment with DNP in T84 and HT-29 cells was prevented by butyrate, subsequently reducing bacterial transcytosis induced by metabolic stress.

A study assessing the interaction between microbiota and host using the mucosa–luminal interface in newly diagnosed pediatric CD patients uncovered that the relative abundance of H_2_S microbial producers is increased, while butyrate producers are decreased [85], which does not support a pathogenic role of butyrate in IBD. These authors also found that the proteomic profile of CD patients was altered, mainly by down-regulation of most mitochondrial proteins, including sulfur dioxygenase (ETHE1), thiosulfate sulfurtransferase (TST) and sulfide quinone oxidoreductase (SQRDL), as well as proteins from the mitochondrial respiratory chain. Thus, according to this study, Crohn´s disease is characterized by an increase in H_2_S microbial producers and down-regulation of H_2_S-detoxifying mechanisms in the host, as well as a decrease in butyrate producers. Butyrate is both an energy source for colonocytes and an activator of genes involved in H_2_S-detoxifying mechanisms. The authors hypothesized that an initial decrease in butyrate producers may dampen H_2_S-detoxifying mechanisms, making the host more susceptible to damage produced by an increased abundance of H_2_S producers. The abundance of butyrate has also been reported by Smith et al. to regulate fuel utilization by epithelial mitochondria [86]. Both long- and short-chain fatty acids are oxidized in the colon, although in an inverse relationship. Butyrate induces the oxidation of short-chain fatty acids while inhibiting the oxidation of long-chain fatty acids, which highlights the interlink between bacterial metabolism and epithelial mitochondrial function. The authors further observed that mice with colonic inflammation induced by *C. rodentium* inhibits oxidation of both long- and SCFA and alters mitochondrial structure, number and position within epithelial cells. SCFA are partly generated by fermentation of β-glucans. In two different studies by El-Deeb et al., the oxazolone-induced colitis rat model was treated with β-glucans [87,88]. In both studies, the increased production of short-chain fatty acids was accompanied by a counteracting effect of oxazolone regarding mitochondrial parameters like ATP, citrate synthase activity, mitochondrial membrane potential and mitochondrial ROS. Treatment with β-glucans also rescued the gene expression of *Pgc1α*. Thus, the impact of diet on gut microbiota would have subsequent effects on the metabolic activity of mitochondria within the intestinal mucosa. In fact, a cohort study on pre-IBD patients and healthy controls revealed that the combination of a high-fat diet with antibiotic usage was associated with an 8.6-times higher risk of having pre-IBD [89]. The authors then generated a pre-IBD mouse model (C57BL/6J maintained on a high-fat diet, which received a single dose of streptomycin by oral gavage and one day after were inoculated with strains of *E. coli*. Exposure to pre-IBD risk factors reduced epithelial ATP levels and gene expression of components of the electron transport chain and made mitochondria shift from oxidative phosphorylation to glycolysis. Furthermore, exposure to pre-IBD risk factors down-regulated the expression of *Pgc1α* and *Sirt3* in colonic epithelium. The authors described that administration of 5-amino salicylic acid rescues the gene expression of *Pgc1α* and *Sirt3* and the levels of ATP in the colonic epithelium, suggesting that 5-amino salicylic acid prevents mitochondria from the impact of pre-IBD risk factors. Altogether, these studies show how the metabolic activity of mitochondria in the intestinal epithelial cells is tightly linked to the metabolism of gut bacteria and how this balance is loss during colitis.

### 3.8. Role of Mitochondria in Gut–Brain Axis

Neuropsychiatric conditions like stress, anxiety and depression are frequently found in IBD patients and it is thought to be not only a consequence of the poor quality of life of IBD patients, but a causative, pro-inflammatory factor enhancing disease symptoms and likely playing a role in the development of the disease. In this context, the hippocampus has a critical role in the mechanisms underlying mood disorders, and two different studies have found a link between mitochondrial function in the hippocampus and mood disorders in the context of IBD. In the first study, Haj-Mirzaian et al. performed intrarectal injection of dinitrobenzene sulfonic acid to induce colitis in wild-type male mice, and they observed that induction of colitis was associated with depressive- and anxiety-like behavior and mitochondrial dysfunction in the hippocampus [90]. In a more recent study, Zhang et al. assessed the effect of electroacupuncture and moxibustion in the hippocampus of DSS-treated mice [91]. Interestingly, oral DSS treatment induced a decrease in mitochondrial activity (as measured by mitofilin) in the hippocampus. Electroacupuncture and moxibustion prevented histological damage and blocked the infiltrating inflammatory cells in the mucosa. In the hippocampus, electroacupuncture and moxibustion promoted astrocyte activation and rescued mitochondrial activity to levels similar to those of untreated mice. These two studies show the interlink between colonic inflammation and the hippocampus, more specifically, mitochondrial function in the hippocampus. Further studies are needed to better define the role of mitochondria in the transduction of signals from colon to hippocampus and vice versa.

### 3.9. Role of Mitochondria in Colitis-Associated Colorectal Cancer (CRC)

One of the most concerning effects of IBD is the development of CRC, which is believed to be the consequence of inflammation maintained over a long time. However, inflammation is not the only molecular mechanism governing cell transformation to tumor, and mitochondrial function is also involved in these mechanisms.

Insulin-like growth factor (IGF) regulates energy metabolism, cell growth and cancer. Activation of its receptor, IGF-1R, enhances cell cycle progression and inhibits apoptosis. In a study assessing the role of IGF-1R in colitis-associated CRC, the authors observed that heterozygous *Igf-1r*^+/−^ mice showed attenuated DSS-induced colitis and developed fewer tumors [92]. In fact, heterozygous mice did not develop large-size tumors, which were frequent in DSS-treated wild-type mice. In addition, *Igf-1r*^+/−^ mice showed decreased oxidative stress. Using the *Igf-1r*^+/−^ mice and the human colonic cancer cell line HT-29, the authors observed that heterozygosity of *Igf-1r* and gene silencing of *IGF-1R*, respectively, favored oxidative phosphorylation against glycolysis. Furthermore, HT-29 cells exposed to a low concentration of H_2_O_2_ showed dissipation of mitochondrial membrane potential and mitochondrial fragmentation, while *IGF-1R*-silenced cells maintained a normal mitochondrial shape and membrane potential. Furthermore, the reduction in the levels of mitochondrial fusion proteins in *Igf-1r*^+/−^ mice after treatment with DSS was not as striking as in wild-type mice. All these results suggest a pivotal role of IGF-1R in balancing oxidative stress and mitochondrial dynamics during colitis and tumorigenesis.

A proteomic study compared the protein profiles of colonic epithelium from (i) UC patients without dysplasia; (ii) non-dysplastic colonic tissue from UC patients with high-grade dysplasia or cancer; (iii) high-grade dysplastic tissue from UC; and (iv) normal colons [93]. This study observed that six, eight and twelve mitochondrial proteins were differentially expressed, respectively, by at least two-fold in UC without dysplasia, non-dysplastic tissue from UC progressors and dysplastic tissue. The number of proteins involved in oxidative activity differentially expressed were five, seven and thirteen in each group, respectively. The authors confirmed with histology the over-expression of carbamoyl phosphate synthetase 1 in both dysplastic and non-dysplastic tissue from UC progressors. This protein is an abundant protein in the mitochondrial matrix involved in the urea cycle and the production of nitric oxide. The results from this study could be, however, indicative of different mitochondrial mass in different stages.

Ussakli et al. used the loss of COX as a surrogate marker of mitochondrial dysfunction [94]. They observed that, while patches of COX loss were found occasionally in tissues from non-progressor UC patients, large areas of COX loss were found in non-dysplastic tissue from progressor UC patients, suggesting that mitochondrial dysfunction precedes tumorigenesis. However, the authors found that, while COX loss was evident in non-dysplastic areas adjacent to cancer and areas with low-grade dysplasia, cancer tissues showed increased expression of COX. The mitochondrial loss in pre-neoplasia and regain of mitochondria during cancer was also assessed using levels of mitochondrial DNA and appeared to be driven by the expression of PGC1α. A study published a few years later assessing the mutations in mtDNA in colorectal cancer was in line with these results. Baker et al. observed that mutations in mitochondrial DNA were abundant in biopsies from UC patients and increased in progressor patients [95]. The mutations were randomly distributed through the coding region, but appeared to be over-represented in the D-loop. Mutations in mitochondrial DNA increased in number and pathogenicity in early dysplasia, but decreased in number and pathogenicity in cancer samples, which suggests that functional mitochondria are necessary for malignant transformation. An increase in mitochondrial DNA mutations leading to triggering of dysplasia could be the result of oxidative damage caused during the course of UC. However, the authors of the latter study observed that clonal and subclonal mutations are not significantly represented by C > A conversions, the signature of oxidative damage. Instead, the authors found a C > T conversion rate of eight- to sixteen-fold times more than what would be expected by chance, which is indicative of mtDNA replication errors in the heavy chain.

Another study assessed the mutations in mitochondrial DNA in healthy controls and non-neoplastic and tumor samples from progressor UC patients [96]. Similar to previous studies, the authors observed that mutations in the mitochondrial DNA were more abundant in samples from UC patients compared to healthy controls. However, the authors did not find differences between non-neoplastic and tumor tissues. This observation is again in line with the study by Baker [95] that found that mutations in mitochondrial DNA increase during early dysplasia and decrease in tumor samples.

Mitochondrial function in the context of UC and colitis-induced CRC was also examined by assessing the effects of TFAM knock-down in a DSS-AOM mouse model [97]. The authors of this study observed that knock-down of TFAM increased the susceptibility of DSS-AOM mice to colitis-associated cancer and increased the levels of the pro-inflammatory cytokines IL-1β, IL-6, IL-11 and TNF-α. Furthermore, protein expression of TFAM was down-regulated in biopsies from active CD and UC patients and correlated with C-reactive protein. These findings suggest that down-regulation of TFAM is induced via IL-6/STAT3/miR-23b. In contrast, over-expression of TFAM protected mice from intestinal inflammation and colitis-associated tumorigenesis. However, the authors also found that TFAM was up-regulated in tumors from colitis-associated cancer patients, which again supports the hypothesis that dysfunctional mitochondria promote tumorigenesis, while functional mitochondria are positively selected during tumor growth, likely because of their critical role in cell bioenergetics.

The complex role of mitochondria in colitis-associated cancer, with opposite roles during early dysplasia and tumor growth, makes it difficult to assess the effects of anti-cancer drugs in terms of mitochondrial function. For instance, a study assessed the anti-cancer effect of atractylenolide I, the main biologically active ingredient in *Atractylodes macrocephala*, a medicinal plant. The authors found that atractylenolide I induces apoptosis, increases ROS content and reduces mitochondrial membrane potential in HCT116 and SW480, two CRC cell lines [98]. Atractylenolide I also reduces the number of large tumors (>2 mm) by 30% in the DSS-AOM mouse model. However, the authors did not report the number of pre-neoplastic lesions. According to these results and studies observing the positive selection of functional mitochondria in tumors, it may be suggested that mitochondrial dysfunction induced by atractylenolide I inhibits tumor growth, but no preventive effect was reported or expected according to this hypothesis. Another study using the DSS-AOM mouse model observed that pre-treatment with metformin prevented alteration of mitochondrial morphology induced by DSS-AOM treatment [99], perhaps via activating the LKB1/AMPK pathway. Likely, the reduction in tumor number was a result of the preservation of mitochondrial function during early dysplasia.

In conclusion, mitochondria seem to work as a double-edged sword in colitis-associated cancer (Figure 3), with a pre-cancer stage where colitis activity correlates with mitochondrial dysfunction. Mutations in the mitochondrial genome and mechanisms like PGC1α and TFAM would contribute to the mitochondrial dysfunction leading to tumorigenesis. In a later stage, tumors would positively select functional mitochondria and up-regulate proteins involved in mitochondrial biogenesis (PGC1α) and mitochondrial DNA replication (TFAM) to ensure their bioenergetic demand.

### 3.10. Role of Mitochondria in IBD-Associated Arthritis and Sarcopenia

The only article found associating mitochondrial function with arthritis in IBD patients dates from 1983 [100]. The authors observed that B cells from the synovial membrane contained enlarged, ovate mitochondria with sparse cristae that were frequently compartmentalized and surrounded by fibrillary material.

Similarly, we found only one article in our search linking mitochondrial function with sarcopenia in the context of IBD. The study described that Complex I activity in PBMCs is decreased in malnourished patients, irrespectively of having a diagnosis of IBD [101]. The authors further observed that Complex I activity increased after 1 week of nutritional support, suggesting that mitochondrial function can be rapidly modulated.

### 3.11. Interaction with Drugs

Different drugs (nonspecific or IBD-targeted therapies) have effects on mitochondrial function, which may not only have an impact on disease course, but also predispose treated patients to potential side-effects of the drugs. For instance, both sulfasalazine [102] and mesalazine [103] side-effects on renal injury and cardiotoxicity, respectively, have been related to their effects on mitochondrial function in these tissues. In the first study, kidneys from sulfasalazine-treated rats showed increased oxidative stress (increased ROS and lipid peroxidation and depleted glutathione reservoir). Analysis of the renal mitochondria showed decreased succinate dehydrogenase activity, mitochondrial swelling, mitochondrial depolarization and increased mitochondrial ROS [102]. Similarly, in the second study, the authors observed decreased succinate dehydrogenase activity, increased mitochondrial ROS, mitochondrial swelling and mitochondrial depolarization in hearts from mesalazine-treated rats [103].

NSAIDs have also been repeatedly associated with IBD. A study assessing the effects of NSAIDs on mitochondrial function in intestinal mucosa observed that indomethacin induced uncoupling of mitochondrial respiration and alteration of mitochondrial morphology, with patchy swelling, mitochondrial elongation and loss of cristae [104]. These alterations were accompanied by increased intestinal permeability and ulcers. The authors observed that a combination of aspirin and dinitrophenol and an uncoupling agent yielded similar results, suggesting that alteration of mitochondrial function is necessary for the damaging effect of indomethacin.

Finally, we found two articles reporting mitochondrial benefits of currently approved therapies for IBD. One of these studies assessed the therapeutic effects of infliximab on gut mucosa and observed abnormal, dilated mitochondria in samples obtained before treatment. Interestingly, these mitochondrial defects were corrected after 4 weeks of infliximab treatment [105]. In a previous study, rats treated with metronidazole, an antibiotic commonly used to reduce the risk of post-operatory recurrence, showed reduced oxidative stress (protein carbonyls and malondialdehyde) in the colon [106]. The results of this study suggest that metronidazole may have additional benefits on the colon of IBD patients, reducing oxidative stress.

### 3.12. Mitochondria as a Therapeutic Target

As mitochondrial damage and dysfunction, the generation of ROS and oxidative stress are all being increasingly recognized as hallmarks of IBD, several studies assessing the anti-colitic effects of new drugs and natural compounds include mitochondrial function as an endpoint marker in pre-clinical trials using different animal models. In Table 2, we have summarized the tested compounds, their effects on colitis, their specific effects on mitochondria and the animal model used.

### 3.13. Future Perspectives

Mitochondrial dysfunction and subsequent oxidative stress are being increasingly recognized as disease hallmarks in IBD. As shown in this systematic review, mitochondria are a hub where many different IBD disease mechanisms are interconnected (Figure 4). The literature shows that mitochondrial dysfunction is not only a consequence of the inflammatory microenvironment, but rather a causative agent with an impact on the redox status of intestinal mucosa, integrity of the intestinal barrier, immune system and microbiota–gut and brain–gut axis. Because of this, modulating mitochondrial function is a potential therapeutic target with promising benefits. In fact, a phase 2b randomized placebo-controlled trial of oral MitoQ on UC is being carried out [118]. This trial aims to recruit 200 adults with a confirmed diagnosis of UC (Mayo score). Participants will receive the standard dose of oral prednisolone alongside either MitoQ or placebo for 24 weeks (daily dose). The colonic mucosa, clinical activity, blood stool and calprotectin levels will be assessed at 12 and 24 weeks from the start of the treatment. Primary and secondary endpoints cover clinical aspects of UC (clinical response at 12- and 24-week time-points, endoscopic response and normalization of calprotectin levels, among others). However, it would be interesting to assess mitochondrial features previously found in UC patients, like increased plasma mtDNA levels, markers of oxidative stress, activation of mitochondrial UPR or gene expression of key players of mitochondrial homeostasis like PGC1α or prohibitin. The experimental assessment of these mitochondrial features in combination with the clinical endpoints may serve to confirm the direct impact of MitoQ in mitochondrial function underlying therapeutic benefits in UC.

Mitochondrial dysfunction is being increasingly recognized as a hallmark of IBD by translational scientists, and a better understanding of this mechanism might bring benefits to patients with IBD. However, a more focused attention on mitochondrial function by physicians and the pharmaceutical industry is necessary to translate this knowledge back to the bedside. Thus, as in the clinical trial previously discussed, we consider that assessment of mitochondrial function not only in translational research studies, but also in clinical trials, might be beneficial for the field.

### 3.14. Limitations of This Review

This manuscript revised a total of 117 articles covering the role of mitochondrial function in IBD. The vast majority of these articles have found a clear association of IBD with mitochondrial dysfunction, and the different mechanisms in which mitochondria are linked to IBD are described. However, 29 articles were not included in this review for not being accessible. Furthermore, 14 additional articles were discarded for being in a language other than English. Therefore, we cannot exclude that other mechanisms involving mitochondria in IBD not discussed in this review are already reported in the literature.

## 4. Conclusions

In conclusion, this systematic review highlights the important role of mitochondrial function in IBD and opens avenues to new therapeutic approaches targeting the different mechanisms previously described.

## Figures and Tables

**Figure 1 ijms-24-17124-f001:**
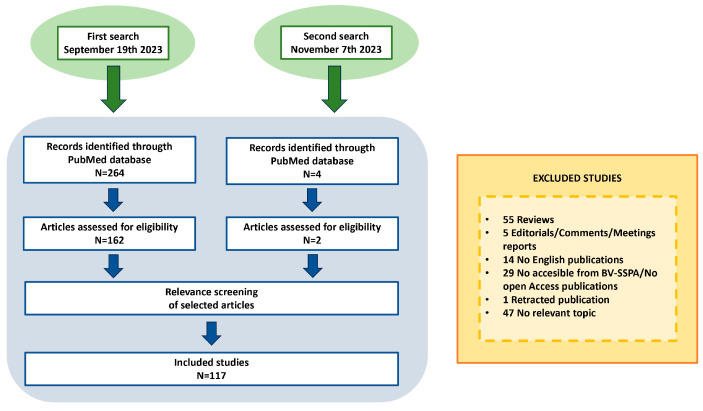
Flow chart of the literature search.

**Figure 2 ijms-24-17124-f002:**
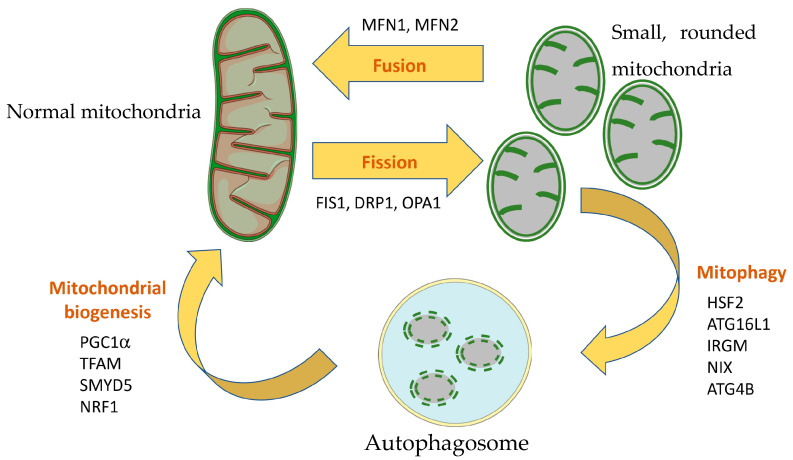
Altered genes of mitochondrial dynamics, biogenesis and mitophagy pathways in IBD.

**Figure 3 ijms-24-17124-f003:**
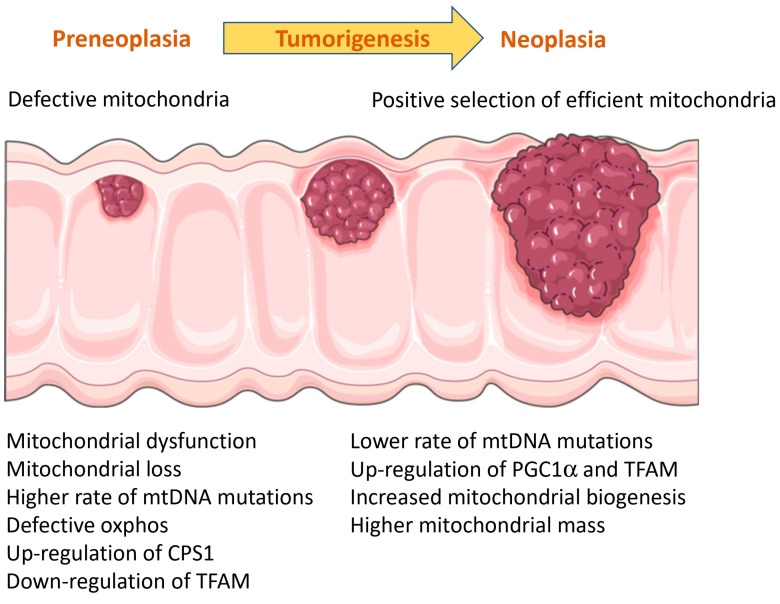
Double-edged behavior of mitochondrial function during different phases of tumorigenesis.

**Figure 4 ijms-24-17124-f004:**
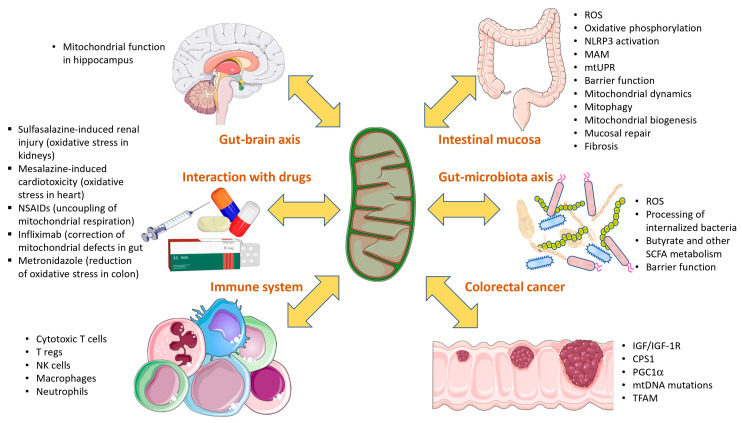
Mitochondria interlinks different pathomechanisms in IBD. Abbreviations: ROS, reactive oxygen species; MAM, mitochondrial–ER-associated membrane; mtUPR, mitochondrial unfolded protein response; SCFA, short-chain fatty acids; mtDNA, mitochondrial DNA; NSAIDs, non-steroidal anti-inflammatory drugs.

**Table 1 ijms-24-17124-t001:** Genes and proteins associated to IBD and involved in mitochondrial function.

Gene/Protein/Signature	Type of Study/Sample	Role in Mitochondria	Ref
OCTN2	GWAS meta-analyses	Carnitine transporters	[2]
*PPIF*	Enhancer screening	Control of the mitochondrial permeability transition and mitochondrial membrane potential	[3]
*MT-ND4*	GWAS	Subunit of the respiratory electron transport chain Complex I	[4]
Down-regulation of nuclear- and mitochondrial-encoded mitochondrial genes, genes from the TCA cycle, and metabolic functions	RNA-seq (rectal samples from pediatric and adult UC)	Mitochondrial function	[5]
Down-regulation of genes involved in mitochondrial respiration in non-inflamed and inflamed tissue from UC and CD	Gene expression microarray (mucosal biopsies from UC, CD, and unclassified IBD	Mitochondrial respiration	[6]
Decreased expression of proteins related to mitochondrial energy metabolism	Proteomics (colonic tissue from DSS-treated rhesus macaques)	Mitochondrial respiration and metabolism of fatty acids	[7]
Increased expression of STEAP4 (a ferrireductase)	Proteomics (colon from DSS-treated mice)	Mitochondrial iron balance	[8]
Up-regulation of SIRT3 and SIRT5; differentially acetylated proteins enrichment in the TCA cycle and fatty acid metabolism	Proteomic analysis of lysine-acetylated proteins and acetylation sites (colon from DSS-treated mice)	TCA cycle and metabolism of fatty acids	[9]
Decreased expression of *PDHA1*, *DBT*, *DLAT*, *LIAS*.	Gene expression meta-analysis of genes involved in cuproptosis	TCA cycle, glycolysis, gluconeogenesis, lipid, pyruvate and propanoate metabolism	[13]

Abbreviations: Ref, reference; GWAS, genome-wide association study; RNA-seq, RNA sequencing; TCA, tricarboxylic acid.

**Table 2 ijms-24-17124-t002:** Natural compounds testing effects in models of colitis and mitochondrial function.

Compound	Natural Source	Model	Benefits on Colitis	Benefits on Mitochondrial Function	Ref.
Asiatic acid	Chinese herb *Centella asiatica*	DSS-induced colitis (C57BL/6J female mice)	- Prevention of weight loss - Reduction in DAI - Attenuation of colon shortening - Reduced pro-inflammatory cytokine profile	- Inhibition of ROS production - Preservation of mitochondrial membrane potential	[107]
Qing Dai powder	Qing Dai herb	NSAID-induced cell injury in RGM1 and IEC6 cell lines	- Reduced cytotoxicity - Prevention of viability loss	- Inhibition of mitochondrial ROS production - Preservation of mitochondrial membrane potential - Prevention of mitochondrial swelling	[108]
FL3 and FL37	Flavaglines found in medicinal plants of Southeast Asia	- Caco2-BBE cell line - IEC6 cell line	- Decreased apoptosis - Prevention of TNFα- or IFNγ-induced NF-κB p65, apoptosis and barrier dysfunction	- Prevention of TNFα-induced decrease in ATP and mitochondrial ROS production - Preservation of Complex I activity	[109]
FL3	Found in medicinal plants of Southeast Asia	DSS-induced colitis (C57BL/6J male mice)	- Prevention of weight loss - Reduction in DAI - Moderate inflammatory infiltration - Reduced crypt loss and ulceration - Reduced histological score	- Reduced oxidative stress (lipid peroxidation and protein carbolnyls)	[109]
Extract of *Spirogyra neglecta*	*Spirogyra neglecta* (freshwater green alga found in Thailand)	DSS-induced colitis (male Crl: CD1 (ICR) mice)	- Amelioration of submucosal edema - Reduced infiltration of inflammatory cells - Reduced loss of crypts, epithelial erosion and ulceration	- Induction of antioxidant enzymes and preservation of mitochondrial function observed in proteomic analysis	[110]
Dried apple peel powder	Apple	DSS-induced colitis (C57BL/6J male mice)	- Prevention of weight loss - Prevention of fecal bleeding - Improved stool consistency - Attenuation of colon shortening - Reduction in lymphocyte infiltration, mucosal erosion, crypt damage and ulcer formation - Reduced pro-inflammatory cytokine profile	- Reduced oxidative stress (malondialdehyde and intramitochondrial H_2_O_2_) - Enhanced β-oxidation - Enhanced ATP production - Partial rescue of PGC1α expression - Preservation of mitochondrial morphology	[111]
Flavonoid VI-16	Synthetic (although flavonoids are present in plants)	DSS-induced colitis (C57BL/6J male mice)	- Attenuation of weight loss - Attenuation of colon shortening - Reduced inflammatoryinfiltration - Preservation of colonic architecture and mucosal damage - Decrease in histological colondamage score - Inhibition of NLRP3 activation and expression of IL-1β and IL-18	- Reduction in mitochondrial ROS production	[112]
Gly-Pro-Ala peptide	Isolated from fish skin gelatin hydrolysate	MODE-K cell line	- Prevention of LPS-induced loss of ZO-1 and occludin	- Activation of mitophagy	[113]
Gly-Pro-Ala peptide	Isolated from fish skin gelatin hydrolysate	DSS-induced colitis (C57BL/6J male mice)	- Attenuation of weight loss - Attenuation of colon shortening - Reduction in DAI - Improved histopathology - Reduced pro-inflammatory cytokine profile - Improved barrier function (higher levels of ZO-1 and occludin)	- Reduced oxidative stress (malondialdehyde and H_2_O_2_)	[113]
Schisandrin B	*Schisandra chinensis*	HCT-116 cell line	- Reduced pro-inflammatory cytokine profile	- Decreased ROS production	[114]
Schisandrin B	*Schisandra chinensis*	DSS-induced colitis (C57BL/6J mice)	- Reduced ulcer area and ulcer score - Attenuation of weight loss - Reduction in DAI - Reduction in macroscopic score - Reduced pro-inflammatory cytokine profile		[114]
Demethyleneberberine	*Coptis* *chinensis* Franch	RAW264.7 cell line	- Reduced levels of IL-1β	- Blocked excessive mitochondrial biosynthesis - Maintained mitochondrial homeostasis during inflammatory response	[115]
Demethyleneberberine	*Coptis* *chinensis* Franch	DSS-induced colitis (C57BL/6J female mice)	- Attenuation of colon shortening - Attenuation of colonic tissue mass score - Reduced neutrophil infiltration - Reduced histological damage - Reduced levels of IL-1β	- Reduced circulating levels of mtDNA	[115]
Licorice	Dry roots and rhizomes of the leguminous plants *Glycyrrhiza uralensis* Fisch, *Glycyrrhiza inflata* Bata or *Glycyrrhiza glabra*	DSS-induced colitis (BALB/C mice)	- Attenuation of weight loss - Attenuation of colon shortening - Reduced histological damage - Reduced pro-inflammatory cytokine profile	- Increased number of mitochondria - Preservation of mitochondrial cristae - Prevention of mitochondrial swelling - Increased mitophagy - Reduced oxidative stress (malondialdehyde) - Reduced ROS production	[116]
Atractylenolide III	Root extracts of *Atractylodes macrocephala* Koidz	DSS-induced colitis (C57BL/6J male mice)	- Prevention of weights loss - Attenuation of colon shortening - Reduction in DAI - Prevention of rectal bleeding - Prevention of barrier dysfunction (increased levels of Zo-1 and Occludin) - Reduced pro-inflammatory cytokine profile	- Reduced oxidative stress (malondialdehyde) - Increased mitochondrial mass (higher levels of mtDNA copy number and Tom20) - Increased Complex I and IV activities - Preservation of mitochondrial biogenesis (increased expression of PGC-1α, NRF-1, NRF-2 and Tfam)	[117]

Abbreviations: Ref: reference; DAI: disease activity index.

## Data Availability

No new data has been created, since this is a systematic review. In any case, figures and tables are original from this review. They are contained within the article.

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
