# Peer review of "Role of Mitochondria in Inflammatory Bowel Diseases: A Systematic Review"

_ijms, 2023, doi:10.3390/ijms242317124_

Round 1
Reviewer 1 Report
Comments and Suggestions for Authors
The paper is comprehensive and well-prepared. It could benefit of adding a diagram/flow chart for the literature search.
Generally, it lacks illustrative materials. Authors could consider one more table for the first data from GWAS, etc.
The paper is more than 40 pages.
Author Response
- The paper is comprehensive and well-prepared. It could benefit of adding a diagram/flow chart for the literature search.
We thank the reviewer for considering that the article is well-prepared. Following the reviewer’s suggestion, we have included additional diagram to explain the literature search (Figure 1).
- Generally, it lacks illustrative materials. Authors could consider one more table for the first data from GWAS, etc.
We thank the reviewer for his suggestion to improve the manuscript. We have prepared a table (Table 1) summarizing the main findings covered in section 2.2 “Omics signatures as evidence of the role of mitochondria in IBD”. In addition, we have added two more figures (Figure 2 and Figure 3) summarizing different sections of the manuscript.
- The paper is more than 40 pages.
The reviewer is right about the length of this article. As opposed to a regular literature review, in a systematic review we have to discuss every article fulfilling the inclusion criteria. In our case, we had to synthesize all the information found in 117 articles. We considered important to include all minimal information to understand every article cited and we aimed to keep the overall readability.

Reviewer 2 Report
Comments and Suggestions for Authors
The manuscript by María JoséSánchez-Quintero et al. summarized the recent advances in the study of the role of mitochondria in IBD. The topic is interesting and I have the following suggestions:
1, in figure 1, the molecules that connects the mitochondria with different diseases and organs must be specified. This figure can be further improved.
2, table 1 should be in a three-line format.
3, future directions and current limitations of the study must be discussed.
4, are there any human clinical studies on the functions of mitochondria in IBD? The authors should discuss.
Author Response
- In figure 1, the molecules that connects the mitochondria with different diseases and organs must be specified. This figure can be further improved.
We agree with the reviewer and we followed his suggestion to improve this figure adding molecules and pathways involved in the link between different topics with IBD.
- Table 1 should be in a three-line format.
We formatted old Table 1 (now Table 2 in the revised section) as a three-line table format.
- Future directions and current limitations of the study must be discussed.
We failed to comment on the limitations of this review and we thank the reviewer for noticing that. We included a section named “limitations of this review” (2.14). Future directions are discussed in the section 2.13, “future perspectives”. However, we included one more paragraph to make a clearer statement of what we consider it should be a future direction.
- Are there any human clinical studies on the functions of mitochondria in IBD? The authors should discuss.
As discussed in the section 2.13, future perspectives, there is an ongoing clinical trial treating patients with MitoQ, an antioxidant targeting mitochondrial ROS. However, this clinical trial does not contemplate any aspect of mitochondrial function as an endpoint criteria, which is also discussed in the section 2.13.

Round 2
Reviewer 2 Report
Comments and Suggestions for Authors
The authors have revised the manuscript accordingly. It can be considered for publication.